# Hyposalivation, oral health, and *Candida* colonization in independent dentate elders

Nada Buranarom[1¤], Orapin Komin[2]*, Oranart Matangkasombut[3,4]*

**1** Graduate Program in Geriatric Dentistry and Special Patient Care, Faculty of Dentistry, Chulalongkorn University, Bangkok, Thailand, **2** Department of Prosthodontics, Faculty of Dentistry, Chulalongkorn University, Bangkok, Thailand, **3** Department of Microbiology and Research Unit on Oral Microbiology and Immunology, Faculty of Dentistry, Chulalongkorn University, Bangkok, Thailand, **4** Laboratory of Biotechnology, Chulabhorn Research Institute, Bangkok, Thailand

¤ Current address: Dental Department, Phaholphonpayuhasena Hospital, Park Phreak, Mueang Kanchanaburi District, Kanchanaburi, Thailand
* oranart.m@chula.ac.th (OM); orapin.geriatric@gmail.com (OK)

**Data Availability Statement:** All relevant data are within the paper and its Supporting Information files.

**Funding:** OM received fundings from Strategic research grant from Faculty of Dentistry and

## Abstract

Hyposalivation is an important problem in elders and could interfere with several oral functions and microbial ecology. While the number of independent elders who retain more natural teeth increases worldwide, few studies examined hyposalivation in this population. Thus, this study aims to examine relationships between hyposalivation, oral health conditions and oral *Candida* colonization in independent dentate elders and evaluate factors associated with salivary flow and *Candida* carriage. We conducted a cross-sectional study in fifty-three dentate elders (≥65 years old with at least 4 pairs of posterior occlusal contacts) with no, or well-controlled, systemic conditions. Participants were interviewed for medical history, subjective dry mouth symptoms, oral hygiene practices and denture information. Unstimulated and stimulated salivary flow rates, objective dry mouth signs, gingival, tongue-coating, and root-caries indices were recorded. Stimulated saliva was cultured on Sabouraud-dextrose agar for *Candida* counts. *Candida* species were identified using chromogenic *Candida* agar and polymerase chain reaction. Statistical significance level was set at $p<0.05$. The results showed that hyposalivation was associated with higher gingival and tongue-coating indices ($p = 0.003$ and $0.015$, respectively), but not root-caries index. Hyposalivation was also associated with higher prevalence of oral *Candida* colonization ($p = 0.010;$ adjusted OR = 4.36, 95% confidence interval = 1.29–14.72). These two indices and *Candida* load were negatively correlated with unstimulated and stimulated salivary flow rates. Interestingly, non-*albicans Candida* species were more prevalent in denture wearers ($p = 0.017$). Hence, hyposalivation is a risk factor for poorer oral health and oral *Candida* colonization in independent dentate elders. Because of its potential adverse effects on oral and systemic health, hyposalivation should be carefully monitored in elders.

Ratchadaphiseksomphot Endowment Fund of Chulalongkorn University (to Research Unit on Oral Microbiology and Immunology). The funders had no role in study design, data collection and analysis, decision to publish, or preparation of the manuscript.

**Competing interests:** The authors have declared that no competing interests exist.

## Introduction

As global life expectancy increases, the ageing population is continually growing worldwide [1]. For example, in Thailand, the number of elders are expected to reach over 20 million, or 30% of the population, by 2035 [2]. Co-existing systemic diseases and the use of multiple medications make elders more vulnerable to oral problems, such as tooth loss, dental caries, periodontitis, dry mouth, oral candidiasis and oral precancer/cancer [3]. Nevertheless, edentulism has declined, and increasing number of elders live independently and retain more natural teeth [4].

Decreased salivary flow or hyposalivation is a common problem in older people [5]. Hyposalivation may result in xerostomia, the subjective feeling of dry mouth, but xerostomia may be experienced in cases with normal salivary gland function [5]. The overall estimated prevalence of dry mouth (xerostomia or hyposalivation) was 22.0% and the prevalence was higher in the elderly population [4, 6]. Although the production of saliva and its composition are largely age-independent in healthy people, a large number of medications can affect salivary flow [7]. Thus, elders may be more prone to hyposalivation. Moreover, higher prevalence of hyposalivation was reported in elders with edentulism or with fewer teeth [8]. However, not many studies examined both xerostomia and hyposalivation, and measured both unstimulated and stimulated salivary flow rates in the same population, especially in dentate elders.

The decrease in saliva flow could disrupt several oral functions. The primary functions of saliva include cleansing and lubricating oral soft and hard tissues, preparation of food for initial digestion, bolus formation for swallowing, modulation of taste perception, facilitation of mastication and phonation, maintenance of oral pH within 6.8–7.2, protecting against acidic challenges from cariogenic bacteria, and promoting remineralization of early carious lesions [9]. Moreover, it maintains the equilibrium of oral microbial ecosystem by its immune components, including antibacterial and antifungal factors, such as histatins, defensins, LL-37 and lysozymes [10, 11]. The composition of the salivary proteome may vary depending on several conditions, such as infection, denture stomatitis, or Sjögren syndrome; these changes, especially of the immune-related components, may affect microbial colonization [12].

When salivary flow is significantly reduced, the oral microbiome is altered [11]. Defects in oral clearance, low salivary pH and changes in salivary compositions lead to microbial dysbiosis and increase the risks of oral diseases, including gingivitis, dental caries and fungal infections [13, 14]. *Candida* is a commensal microorganism in the oral cavity. However, when host immune system is compromised or there are local predisposing conditions, these fungi can cause oral and systemic infection (candidiasis) [15]. Reduced salivary flow could lead to increased *Candida* accumulation, which could elevate the risk of oral candidiasis [11]. Other local factors that predispose to oral candidiasis are poor oral hygiene, ill-fitting dentures, poor denture hygiene, or prolonged denture usage [16]. These conditions are prevalent in the elders and may contribute to risk of *Candida* infection. Furthermore, we previously found that denture use was associated with higher colonization of non-albicans *Candida* species (NACS) in xerostomic post-radiation therapy patients [17]. The NACS, such as *Candida tropicalis*, *Candida glabrata*, *Candida dubliniensis*, *Candida krusei*, and *Candida parapsilosis*, can cause infections that are more difficult to treat and may be resistant to antifungal drugs. In addition, the NACS are frequently found in multispecies colonization and may have inter-species interactions with *C. albicans* [18]. These species were commonly found in patients with underlying systemic conditions, such as head and neck cancer, and HIV infection [19–21]. However, information on hyposalivation and oral *Candida* in dentate elders with no or well-controlled systemic conditions is still limited. Because the commensal oral *Candida* may be a source of infection when the host becomes immunocompromised [22], information regarding oral

carriage of *Candida* and factors that affect colonization is important to evaluate risk of infection in the elderly population.

This cross-sectional study aimed to examine the association of oral health status, as measured by gingival, tongue-coating, and root caries indices, and oral *Candida* species colonization, with hyposalivation in independent dentate elders with no, or well-controlled, systemic conditions. We also evaluated the factors that associate with salivary flow rates, *Candida* colonization, and oral health indices.

## Materials and methods

### Study population

Study participants were recruited from 120 elderly dental patients in the waiting list of graduate geriatric clinic at the Faculty of Dentistry, Chulalongkorn University in Bangkok and 240 members of Phaholpolpayuhasena hospital elderly club in Kanchanaburi Province, Thailand. Preliminary screening by reviews of medical/dental records, interview and/or oral examination was performed to identify patients who fit the following eligibility criteria (Table 1).

A total of 53 participants qualified the eligibility criteria, gave written informed consents, and were enrolled in the study. The study protocol was approved by the research ethics committee of the Faculty of Dentistry, Chulalongkorn University (HREC-DCU 2017–094, Approval date: March 2nd, 2018) and Phaholpolpayuhasena hospital (IRB number 2018–01, Approval date: January 16th, 2018) in accordance with the Declaration of Helsinki. The study was conducted during March 2018-March 2019.

Sample size was calculated based on estimated average logCFU/ml and SD values of *Candida* in patients with normal and hyposalivation from a previous study with $\alpha = 0.05$ and power of 90% [23]. The number of samples per group required was 21.

### Data collection

Participants were interviewed for demographic data, including age, underlying medical diseases, xerostomia-inducing drug used [7], oral hygiene practice, denture use and denture hygiene practice. Medical records were also reviewed for medical history. Subjective dry mouth symptoms were obtained by interviewing the subjects using a previously described

**Table 1. Eligibility criteria.**

| Inclusion criteria | Exclusion criteria |
|---|---|
| 1. age 65 years old or over | 1. had used systemic antibiotics or antifungal drugs within the last 6 months |
| 2. at least 4 pairs of posterior occlusal contacts | 2. used topical antibiotics, topical antifungal or topical steroid in the oral cavity within the previous 7 days |
| 3. were in good general health with no or well-controlled systemic conditions (ASA class I or II) | 3. smoked or had history of smoking in the last 5 years |
| 4. willing to participate and able to provide saliva samples. | 4. had acute illness |
| | 5. had poorly controlled systemic disease |
| | 6. had any evidence of having the following conditions: |
| | • precancerous or cancerous oral lesions, |
| | • periodontal pockets deeper than 4 mm (mild gingivitis was acceptable), |
| | • infections related to carious teeth (apical abscess, space infections) |
| | • oral candidiasis. |

questionnaire with minor modifications [5]. Eight questions were included as follow: 1. Does your mouth feel dry at night or on awakening?; 2. Does your mouth feel dry at other times of the day?; 3. Do you keep a glass of water by your bed?; 4. Do you sip liquids to aid in swallowing dry foods?; 5. Does your mouth feel dry when eating a meal?; 6. Do you chew gum daily to relieve oral dryness?; 7. Do you use hard candies or mints daily to relieve oral dryness?; 8. Does the amount of saliva in your mouth seem to be too little? Participants who gave at least 1 positive response to these questions were considered as subject with dry mouth symptoms.

## Oral examination

Information regarding objective dry mouth signs [24], gingival index (GI) [25], tongue-coating index (TCI) [26], root caries index (RCI) [14] were determined upon oral examination by a trained dentist.

Objective dry mouth signs were examined as described [24]. The ten signs of dry mouth include sticking of an intraoral mirror to the buccal mucosa or tongue, frothy saliva, no saliva pooling in floor of mouth, loss of papillae of the tongue dorsum, altered/smooth gingival architecture, glassy appearance of the oral mucosa, lobulated/deeply fissured tongue, cervical caries, and mucosal debris on palate. Participants with at least 1 sign were considered as subject with dry mouth signs.

Gingival index was scored as previously described [25]. Bleeding is assessed upon probing gently along the wall of soft tissue of the gingival sulcus with a periodontal probe at four sites (mesial, distal, buccal and lingual surfaces) of six selected teeth (right maxillary first molar, right maxillary lateral incisor, left maxillary first molar, left mandibular first molar, left mandibular lateral incisor and right mandibular first molar). The scores were from 0 = no inflammation 1 = mild inflammation; 2 = moderate inflammation, and 3 = severe inflammation.

Tongue-coating index was scored and calculated as described [26]. The tongue surface was divided into nine sections and the tongue coating status (0 = coating not visible, 1 = thin coating, 2 = thick coating) was visually scored in each section and combined. The combined value was calculated into percentage of maximum score.

Root caries index was scored as described [14]. All teeth with gingival recession were examined on four surfaces (mesial, distal, buccal and lingual) for root caries using the following criteria: 1. The lesion should be located at the cementoenamel junction or completely on the root surface; 2. There should be a discrete, well-defined, softened area indicating decay; 3. The explorer should enter easily and display some resistance to withdrawal. Restored lesions are counted as root caries only if it is obvious that the lesion originated at the cementoenamel junction. Crowned teeth were not included because the type of lesion that existed prior to the placement of the restoration could not be determined. The percentage of decayed and filled surfaces relative to total number of root surfaces was calculated.

Denture plaque index was scored as previously described [27]. The dentures were rinsed through running tap water, and then painted with erythrosin dye. Excess dye was gently rinsed off after 30 seconds. Eight areas, four on the tissue surface and four on the polished surface were scored for plaque and stain accumulations according to these criteria: 0 = No plaque; 1 = Light plaque, 1% to 25% of area covered; 2 = Moderate plaque, 26% to 50% of area covered; 3 = Heavy plaque, 51% to 75% of area covered; 4 = Very heavy plaque, 76% to 100% of area covered. Denture plaque index was calculated from the sum of eight scores divided by 32. In participants with upper and lower dentures, the average of both denture plaque index scores was used.

## Saliva collection

Participants were instructed not to use any mouth rinse for 12 hours and to withhold oral intake (food, medication, water) and tooth brushing for at least 90 minutes prior to saliva

collection. Saliva specimens were collected between 9:00–11:00 a.m. to minimize variations associated with the circadian cycle. Before saliva collection, subjects were instructed to swallow to clear the mouth from any accumulated saliva. During the collection, participants sat straight with head slightly tilted forward and abstained from speaking and swallowing. Unstimulated whole saliva was collected by spitting the fluid available in the mouth into a graduated sterile tube every 30 seconds for 10 minutes. After 2 minutes break, stimulated whole saliva was collected after chewing a piece of paraffin wax (5x5 cm.) for a period of 2 minutes, then subjects discarded the saliva available in the mouth. Subject continued chewing through the process and spat saliva into a graduated sterile tube every 30 seconds for 5 minutes. The volume of clear saliva was measured to estimate salivary flow rate. Unstimulated salivary flow rate (USFR) of less than 0.1 mL per minute or stimulated salivary flow rate (SFR) of less than 0.7 mL per minute were considered as hyposalivation [28–30].

### *Candida* counts and species identification

Saliva samples were immediately placed on ice and transferred to the laboratory for culture within 3 hours. Each sample was serially diluted to obtain 1:10, 1:100 and 1:1000 dilutions. A volume of 100μl of each dilution was spread on Sabouraud dextrose agar plate containing 5 mg/ml streptomycin and 2500 unit/ml penicillin G sodium and incubated at 37˚C for 48 hours. The number of Colony Forming Unit (CFU) per milliliter of saliva was calculated and log transformed for statistical analyses. Plates without fungal growth at 48 hours were further incubated for up to 2 weeks before being considered as negative.

Ten isolated yeast colonies on Sabouraud dextrose agar per sample were chosen and streaked on chromogenic *Candida* agar (oxoid, UK). *Candida* colonies were initially characterized based on colony colour according to the manufacturer's recommendation (*C. albicans*: green *C. dubliniensis*: green, *C. tropicalis*: metallic blues, *C. krusei*: pink, fuzzy, *C. glabrata*: white to mauve, *C. parapsilosis*: white to mauve). Further species identification were accomplished using polymerase chain reaction (PCR) with species-specific primers as previously described [21, 31]: *C. albicans* (CAL5-NL4CAL, CALB1F-CALB2R), *C. dubliniensis* (CDU2-NL4CAL, DUBF-DUBR), *C. glabrata* (CGL1-NL4CGL1), *C. parapsilosis* (CP4-NL4LEL1), and *C. tropicalis* (CTR22-NLN4CTR).

### Statistical analysis

Demographic data and prevalence of *Candida* species were evaluated by using descriptive statistics. Factors associated with salivary flow rate and *Candida* species colonization were analyzed using Pearson Chi-square test or Fisher's exact test for categorical data, and T-test or Mann-Whitney U test for continuous data as specified. Logistic regression was used to calculate odds ratio adjusted for the effect of age. Correlations among factors were evaluated by Spearman correlation coefficient analysis. All analyses were performed with IBM SPSS statistics version 22. A p-value of less than 0.05 was considered statistically significant.

## Results

### Characteristics of study population

A total of 53 participants were included in this study. The characteristics of the participants are shown in Table 2. The average age was 71.9±6 years. Mean unstimulated and stimulated whole salivary flow rates were 0.35±0.26 ml/minute and 0.97±0.60 ml/minute, respectively. Among the 53 participants, 22 (41.5%) had hyposalivation. The majority of the subjects were female (84.9%). Thirty-four subjects (64.2%) had underlying medical conditions, while 36 subjects

**Table 2. Characteristics of the study population.**

| Variables | Study population (N = 53) | Normal salivation group (N = 31) | Hyposalivation group (N = 22) | Between-group comparisons |
|---|---|---|---|---|
| | Mean±SD (Min-Max) | Mean±SD (Min-Max) | Mean±SD (Min-Max) | p-value |
| **Age (years)** | 71.9±6 (65–92) | 70.5±6.1 (65–92) | 74.0±5.2 (67–83) | 0.009[M]* |
| **Salivary flow rate (ml/min)** | | | | |
| • Unstimulated saliva | 0.35±0.26 (0.05–1.20) | 0.46±0.27 (0.15–1.20) | 0.18±0.11 (0.05–0.40) | <0.001[M]* |
| • Stimulated saliva | 0.97±0.60 (0.10–3.00) | 1.33±0.53 (0.70–3.00) | 0.47±0.20 (0.10–0.85) | <0.001[M]* |
| **Number of remaining teeth** | 25.4±2.7 (16–32) | 25.4±3.2 (16–32) | 25.4±1.9 (20–28) | 0.482 |
| | N (%) | N (%) | N (%) | |
| **Gender** | | | | |
| • Male | 8 (15.1) | 5 (16.1) | 3 (13.63) | 1.000[F] |
| • Female | 45 (84.9) | 26 (83.9) | 19 (86.36) | |
| **Systemic conditions** | | | | |
| • Cardiovascular diseases | 22 (41.5) | 11 (35.5) | 11 (50) | 0.291 |
| • Dyslipidemia | 8 (15.1) | 3 (9.7) | 5 (22.7) | 0.253[F] |
| • Chronic kidney diseases | 5 (9.4) | 2 (6.4) | 3 (13.6) | 0.638[F] |
| • Diabetes mellitus | 4 (7.5) | 3 (9.7) | 1 (4.5) | 0.633[F] |
| • Osteoporosis | 3 (5.6) | 3 (9.7) | 0 (0) | 0.258[F] |
| • Depressive disorders | 3 (5.6) | 1 (3.2) | 2 (9.1) | 0.563[F] |
| • Cerebrovascular diseases | 2 (3.7) | 0 (0) | 2 (9.1) | 0.168[F] |
| • Osteoarthritis | 2 (3.7) | 2 (6.4) | 0 (0) | 0.505[F] |
| • Spondylolisthesis | 2 (3.7) | 0 (0) | 2 (9.1) | 0.168[F] |
| • Parkinson's disease | 2 (3.7) | 0 (0) | 2 (9.1) | 0.168[F] |
| • No underlying conditions | 19 (35.8) | 14 (32.4) | 5 (22.7) | 0.093 |
| **Xerostomic drug use** | | | | |
| • Yes | 36 (67.9) | 18 (58.1) | 18 (81.8) | 0.068 |
| • No | 17 (32.1) | 13 (41.9) | 4 (18.2) | |
| **Brushing after meal** | | | | |
| • Yes | 25 (47.2) | 13 (42.9) | 12 (54.5) | 0.365 |
| • No | 28 (52.8) | 18 (58.1) | 10 (45.5) | |
| **Denture use** | | | | |
| • Yes | 11 (20.8) | 7 (22.6) | 4 (18.2) | 0.745[F] |
| • No | 42 (79.2) | 24 (77.4) | 18 (81.8) | |
| **Subjective dry mouth symptoms** | | | | |
| • Yes | 38 (71.7) | 21 (67.7) | 17 (77.3) | 0.448 |
| • No | 15 (28.3) | 10 (32.3) | 5 (22.7) | |
| **Objective dry mouth signs** | | | | |
| • Yes | 19 (35.8) | 6 (19.4) | 13 (59.1) | 0.003* |
| • No | 34 (64.2) | 25 (80.6) | 9 (40.9) | |

[M] Mann-Whitney U test

[F] Fisher's Exact Test, otherwise Pearson Chi-Square test

*Statistically significant difference (p<0.05)

(67.9%) used xerostomia-inducing drugs. Twenty-five subjects (47.2%) brushed after meals regularly. Eleven subjects (20.8%) wore acrylic removable partial dentures. Thirty-eight subjects (71.7%) and nineteen subjects (35.8%) had dry mouth symptoms and objective dry mouth signs, respectively. Most of the subjects with dry mouth symptoms took frequent sips of water to ease their symptoms; 35 subjects (92.01%) took sips to aid in swallowing dry foods and 30 subjects (78.95%) woke up at night to take sips of water.

**Table 3. Association of oral *Candida* colonization and salivation status.**

| *Candida* species | Study population (N = 53) | *Candida* carriers (N = 25) | Normal salivation group (N = 31) | Hyposalivation group | Between-group |
|---|---|---|---|---|---|
| | N (%) | N (%) | N (%) | (N = 22) | p-value |
| | | | | N (%) | |
| *Candida spp.* | 25 (47.2) | 25 (100) | 10 (32.3) | 15 (68.2) | 0.010* |
| *C. albicans* | 19 (35.8) | 19 (76) | 8 (25.8) | 11 (50.0) | 0.07 |
| Multispecies | 9 (17) | 9 (36) | 3 (9.7) | 6 (27.27) | 0.140[F] |
| Non-*albicans* species | 13 (24.5) | 13 (52) | 5 (16.1) | 8 (36.4) | 0.092 |
| *C. glabrata* | 5 (9.4) | 5 (20) | 2 (6.5) | 3 (13.6) | 0.638 [F] |
| *C. dubliniensis* | 4 (7.5) | 4 (16) | 1 (3.2) | 3 (13.6) | 0.295 [F] |
| *C. parapsilosis* | 4 (7.5) | 4 (16) | 0 (0) | 4 (18.18) | 0.025* [F] |
| *C. krusei* | 2 (3.8) | 2 (8) | 2 (6.5) | 0 (0) | 0.505 [F] |
| *C. tropicalis* | 1 (1.9) | 1 (4) | 1 (3.2) | 0 (0) | 1.000 [F] |

[F] Fisher's Exact Test, otherwise Pearson Chi-Square test

*Statistically significant difference (p<0.05)

There was a statistically significant difference in the age of the participants with and without hyposalivation (p = 0.009). Although hyposalivation was defined as having either low unstimulated or low stimulated salivary flow rates, both were found to be significantly lower in participants classified as having hyposalivation (p<0.001). The prevalence of objective dry mouth signs was greater in the hyposalivation group (p = 0.003), but no difference was observed for subjective dry mouth symptoms (p = 0.448). There was no statistically significant difference in the percentage of xerostomia-inducing drug use, brushing after meal, acrylic removable partial denture used and other medical conditions between the two groups.

## Oral *Candida* species colonization

The prevalence of oral *Candida* species is shown in Table 3. Overall, 25 participants (47.2%) were *Candida* carriers. *C. albicans* was the most commonly detected species (76% of *Candida* carriers), while non-*albicans* species were detected in 52% of *Candida* carriers. Colonization by multiple species (multispecies) was detected in 36% of *Candida* carriers. *C. glabrata* was the most common non-*albicans Candida* species detected (20% of *Candida* carriers), followed by *C. dubliniensis*, *C. parapsilosis*, *C. krusei* and *C. tropicalis* (16%, 16%, 8%, and 4% of *Candida* carriers, respectively). When compared between normal salivation and hyposalivation groups, we found significantly higher *Candida* colonization in hyposalivation group (68.2%) than the normal salivation group (32.3%) (p = 0.010), with odds ratio of 4.50 (95% confidence interval = 1.395–14.52, p = 0.012). Since there was a significant difference in the age of participants in the hyposalivation and normal salivation groups, we analyzed for the effect of age in logistic regression. Hyposalivation was still associated with higher prevalence of *Candida* colonization when controlled for age with adjusted odds ratio of 4.36 (95% confidence interval = 1.29–14.72, p = 0.018). There was no statistically significant difference in the prevalence of multispecies or non-*albicans Candida* species between groups, except for *C.parapsilosis* (p = 0.025).

## Factors associated with oral health status

Oral health status of the study population was evaluated by oral examination and measurements of the gingival index (GI), tongue-coating index (TCI), and root caries index (RCI) (Table 4). Participants with hyposalivation had significantly higher mean GI and TCI (p = 0.003 and 0.015 respectively), but not RCI (p = 0.986). Likewise, participants with

**Table 4. Association of clinical parameters and oral health indices.**

| | N | Gingival index | | Tongue-coating index | | Root caries index | |
|---|---|---|---|---|---|---|---|
| | | Mean±SD | P-value | Mean±SD | P-value | Mean±SD | P-value |
| **Salivation status** | | | | | | | |
| • Normal | 31 | 0.96±0.24 | 0.003* | 13.35±13.73 | 0.015* | 10.28±5.98 | 0.986# |
| • Hyposalivation | 22 | 1.25±0.40 | | 28.28±22.02 | | 10.32±5.02 | |
| **Subjective dry mouth symptoms** | | | | | | | |
| • Yes | 38 | 1.11±0.36 | 0.452 | 21.63±18.89 | 0.096 | 10.78±5.46 | 0.330# |
| • No | 15 | 0.99±0.29 | | 14.26±18.75 | | 9.06±5.76 | |
| **Objective dry mouth signs** | | | | | | | |
| • Yes | 19 | 1.25±0.41 | 0.012* | 22.51±18.09 | 0.307 | 9.61±4.56 | 0.479# |
| • No | 34 | 0.99±0.27 | | 17.89±19.50 | | 10.67±6.06 | |
| **Brushing after meal** | | | | | | | |
| • Yes | 25 | 1.05±0.27 | 0.724 | 19.78±18.57 | 0.634 | 10.58±6.61 | 0.728# |
| • No | 28 | 1.11±0.40 | | 19.34±19.63 | | 10.03±4.51 | |
| **Denture use** | | | | | | | |
| • Yes | 11 | 1.02±0.14 | 0.832 | 22.22±16.10 | 0.298 | 13.33±5.45 | 0.054# |
| • No | 42 | 1.09±0.38 | | 18.84±19.76 | | 9.49±5.35 | |
| ***Candida* spp.** | | | | | | | |
| • Yes | 25 | 1.16±0.38 | 0.169 | 21.89±19.78 | 0.305 | 10.51±5.34 | 0.789# |
| • No | 28 | 1.00±0.30 | | 17.46±18.31 | | 10.09±5.82 | |
| **Non-*albicans*** | | | | | | | |
| • Yes | 13 | 1.19±0.46 | 0.294 | 21.37±18.68 | 0.471 | 12.07±5.32 | 0.184# |
| • No | 40 | 1.04±0.30 | | 18.96±19.25 | | 9.71±5.56 | |
| **Multiple species** | | | | | | | |
| • Yes | 9 | 1.13±0.22 | 0.427 | 20.98±17.52 | 0.526 | 13.05±5.73 | 0.137# |
| • No | 44 | 1.07±0.37 | | 19.25±19.43 | | 9.72±5.40 | |

#Independent t-test, otherwise Mann-Whitney U test

*Statistically significant difference (p<0.05)

objective dry mouth signs had significantly higher GI that those without the signs (p = 0.012). However, the presence of objective dry mouth signs was not associated with TCI (p = 0.307) nor RCI (p = 0.479). The presence of *Candida* was not associated with any of the indices. Interestingly, none of the factors examined had significant relationship with RCI. Nevertheless, participants who wear dentures tended to have higher RCI with marginally significant difference (p = 0.054).

In addition, as shown in Fig 1, we found significant negative correlations between GI and USFR (r = -0.387, p = 0.004) and also between GI and SFR (r = -0.371, p = 0.006) (Fig 1C and 1D). Moreover, there were significant negative correlations between TCI and USFR (r = -0.271, p = 0.049) and between TCI and SFR (r = -0.359, p = 0.008) (Fig 1E and 1F). However, no correlation was observed between RCI and salivary flow rates (Fig 1G and 1H). These findings suggested that high GI and TCI correlate with low unstimulated and stimulated salivary flow rates.

## Factors associated with *Candida* colonization and *Candida* counts

We examined the factors that may associate with risk of *Candida* carriage (Table 5). *Candida* colonization was higher in participants with objective dry mouth signs (p = 0.021) and hyposalivation (p = 0.010). In contrast, gender, xerostomic drug use, subjective dry mouth symptoms,

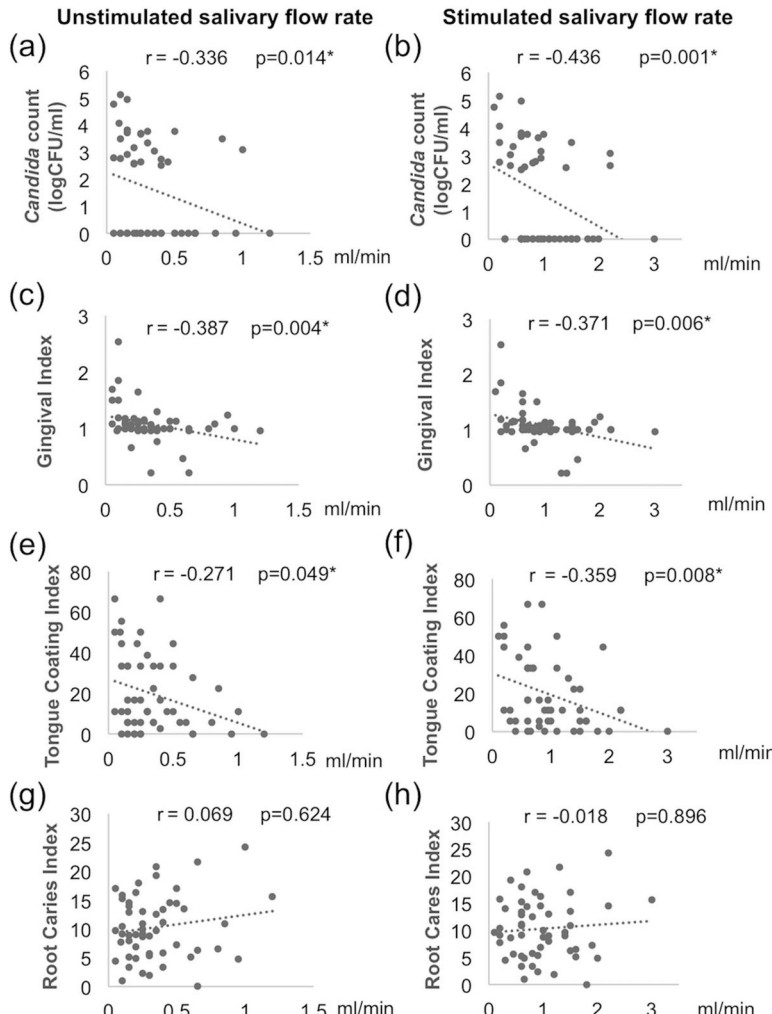

**Fig 1. Correlation between salivary flow rates, *Candida* colonization, and oral health indices.** (a, c, e, g for unstimulated and b, d, f, h for stimulated salivary flow rates) The correlations were shown for (a, b) quantity of *Candida* colonization, (c, d) gingival index, (e, f) tongue-coating index, and (g, h) root caries index. Data were analyzed with Spearman correlation coefficient analysis. The r and p-value of each correlation are shown.

brushing, denture use, nor systemic conditions did not show statistically significant difference. Interestingly, denture use was associated with higher prevalence of non-*albicans Candida* colonization (p = 0.017).

Since *Candida* species can form biofilm on denture surfaces, denture hygiene is likely an important factor that could affect *Candida* colonization in denture wearers. We therefore examined the relationship between denture plaque index and *Candida* colonization, however, we did not detect any significant association (Table 6). Among the denture wearers (N = 11), the mean age of dentures was 5.27±3.00 years old; only 3 (27.3%) wore dentures at night; and all used toothbrush and toothpaste to clean their dentures twice daily. There was no association between night time wearing and *Candida* colonization.

When we examined the quantity of *Candida* colonization among *Candida* carriers (Table 7), we found that participants with subjective dry mouth symptoms had significantly higher number of *Candida* in the saliva than those without the symptoms (3.55±0.75 vs. 2.82

**Table 5. Association of clinical parameters and *Candida* colonization (N = 53).**

| | N | *Candida* | | Non-*albicans* | | Multispecies | |
|---|---|---|---|---|---|---|---|
| | | N = 25(%) | P-value | N = 13(%) | P-value | N = 9(%) | P-value |
| **Gender** | | | | | | | |
| • Male | 8 | 3 (12.00) | 0.708[F] | 0 (0) | 0.176[F] | 0 (0) | 0.324[F] |
| • Female | 45 | 22 (88.00) | | 13 (100.00) | | 9 (100.00) | |
| **Xerostomic drug use** | | | | | | | |
| • Yes | 36 | 18 (72.00) | 0.548 | 7 (53.85) | 0.306[F] | 5 (55.55) | 0.445[F] |
| • No | 17 | 7 (28.00) | | 6 (46.15) | | 4 (44.45) | |
| **Subjective dry mouth symptoms** | | | | | | | |
| • Yes | 38 | 20 (80.00) | 0.205 | 10 (76.92) | 0.736[F] | 8 (88.89) | 0.418[F] |
| • No | 15 | 5 (20.00) | | 3 (23.08) | | 1 (11.11) | |
| **Objective dry mouth signs** | | | | | | | |
| • Yes | 19 | 13 (52.00) | 0.021* | 7 (53.85) | 0.183[F] | 5 (55.55) | 0.255[F] |
| • No | 34 | 12 (48.00) | | 6 (46.15) | | 4 (44.45) | |
| **Salivation status** | | | | | | | |
| • Normal | 31 | 10 (40.00) | 0.010* | 5 (38.46) | 0.092 | 3 (33.33) | 0.140[F] |
| • Hyposalivation | 22 | 15 (60.00) | | 8 (61.54) | | 6 (66.67) | |
| **Brushing after meal** | | | | | | | |
| • Yes | 25 | 14 (56.00) | 0.224 | 7 (53.85) | 0.579[F] | 5 (55.55) | 0.719[F] |
| • No | 28 | 11 (44.00) | | 6 (46.15) | | 4 (44.45) | |
| **Denture use** | | | | | | | |
| • Yes | 11 | 8 (32.00) | 0.056 | 6 (46.15) | 0.017*[F] | 4 (44.45) | 0.076[F] |
| • No | 42 | 17 (68.00) | | 7 (53.85) | | 5 (55.55) | |
| **Systemic conditions** | | | | | | | |
| • Yes | 34 | 17 (68.00) | 0.581 | 6 (46.15) | 0.183[F] | 4 (44.45) | 0.255[F] |
| • No | 19 | 8 (32.00) | | 7 (53.85) | | 5 (55.55) | |

[F] Fisher's Exact Test, otherwise Pearson Chi-Square test

*Statistically significant difference (p<0.05)

±0.37 logCFU/ml, p = 0.025). Participants who used xerostomic drugs, and those who used dentures tended to have higher number of *Candida*, but the difference was not statistically significant. (p = 0.173 and 0.091, respectively)

Although categorical salivation status (normal vs hyposalivation) did not show significant association with the quantity of *Candida* colonization (Table 7), we observed statistically significant negative correlations between *Candida* count and unstimulated salivary flow rate

**Table 6. Association of denture plaque index and *Candida*.**

| | N (%) | Denture plaque index, Mean±SD | p-value |
|---|---|---|---|
| ***Candida* spp.** | | | |
| • Yes | 8 (72.73) | 36.93±23.05 | 1 |
| • No | 3 (27.27) | 33.33±16.04 | |
| **Non-*albicans* Candida** | | | |
| • Yes | 6 (54.55) | 32.55±22.63 | 0.465 |
| • No | 5 (45.45) | 40.04±19.67 | |

Mann-Whitney U test

**Table 7. Association of clinical parameters and quantity of *Candida* colonization (logCFU/ml) among *Candida* carriers (N = 25).**

| | | *Candida* counts (logCFU/ml) | |
|---|---|---|---|
| | N | Mean±SD | P-value |
| **Gender** | | | |
| • Male | 3 | 2.90±0.39 | 0.181 |
| • Female | 22 | 3.47±0.76 | |
| **Xerostomic drug use** | | | |
| • Yes | 18 | 3.54±0.79 | 0.173 |
| • No | 7 | 3.05±0.50 | |
| **Subjective dry mouth symptoms** | | | |
| • Yes | 20 | 3.55±0.75 | 0.025* |
| • No | 5 | 2.82±0.37 | |
| **Objective dry mouth signs** | | | |
| • Yes | 13 | 3.65±0.87 | 0.182 |
| • No | 12 | 3.14±0.52 | |
| **Salivation status** | | | |
| • Normal | 10 | 3.18±0.46 | 0.36 |
| • Hyposalivation | 15 | 3.55±0.87 | |
| **Brushing after meal** | | | |
| • Yes | 14 | 3.17±0.68 | 0.125 |
| • No | 11 | 3.59±0.77 | |
| **Denture use** | | | |
| • Yes | 8 | 3.81±0.86 | 0.091 |
| • No | 17 | 3.21±0.63 | |
| *C. albicans* | | | |
| • Yes | 19 | 3.47±0.73 | 0.265 |
| • No | 6 | 3.19±0.83 | |
| **Non-*albicans*** | | | |
| • Yes | 13 | 3.35±0.82 | 0.479 |
| • No | 12 | 3.46±0.70 | |
| **Multiple species** | | | |
| • Yes | 9 | 3.54±0.90 | 0.734 |
| • No | 16 | 3.33±0.66 | |

Mann-Whitney U test

*Statistically significant difference (p<0.05)

(USFR) (r = -0.336, p = 0.014) and between *Candida* count and stimulated salivary flow rate (SFR) (r = -0.436, p = 0.001) (Fig 1A and 1B). These findings suggested that low unstimulated and stimulated salivary flow rates correlate with higher amounts of *Candida* colonization.

## Factors associated with salivary flow rates

We also examined the factors that associate with salivary flow rates (Table 8). Mean USFR and mean SFR in participants with objective dry mouth signs (0.24±0.25 ml/min and 0.72±0.39 ml/min, respectively, p = 0.003) were significantly lower than those without (0.41±0.25 ml/min and 1.12±0.55 ml/min, respectively, p = 0.003). In contrast, no statistical difference in mean USFR and SFR was observed between participants with subjective dry mouth symptoms and those without (p = 0.118 and 0.188, respectively). Furthermore, mean USFR and SFR of

**Table 8. Association of population characteristics and salivary flow rate.**

| | Unstimulated salivary flow rate (ml/min) | | | Stimulated salivary flow rate (ml/min) | | |
|---|---|---|---|---|---|---|
| | N | Mean±SD | P-value | N | Mean±SD | P-value |
| **Gender** | | | | | | |
| • Male | 8 | 0.33±0.21 | 0.891 | 8 | 1.15±0.56 | 0.262 |
| • Female | 45 | 0.34±0.27 | | 45 | 0.94±0.61 | |
| **Xerostomic drug use** | | | | | | |
| • Yes | 36 | 0.34±0.27 | 0.485 | 36 | 0.89±0.61 | 0.053 |
| • No | 17 | 0.36±0.24 | | 17 | 1.15±0.56 | |
| **Subjective dry mouth symptoms** | | | | | | |
| • Yes | 38 | 0.31±0.25 | 0.118 | 38 | 0.92±0.62 | 0.188 |
| • No | 15 | 0.42±0.27 | | 15 | 1.12±0.56 | |
| **Objective dry mouth signs** | | | | | | |
| • Yes | 19 | 0.24±0.25 | 0.003* | 19 | 0.72±0.39 | 0.003* |
| • No | 34 | 0.41±0.25 | | 34 | 1.12±0.55 | |
| **Denture use** | | | | | | |
| • Yes | 11 | 0.38±0.29 | 0.628 | 11 | 0.93±0.54 | 1 |
| • No | 42 | 0.34±0.25 | | 42 | 0.99±0.62 | |
| ***Candida* spp.** | | | | | | |
| • Yes | 25 | 0.27±0.23 | 0.042* | 25 | 0.77±0.56 | 0.007* |
| • No | 28 | 0.40±0.27 | | 28 | 1.16±0.59 | |
| **Non-*albicans*** | | | | | | |
| • Yes | 13 | 0.32±0.29 | 0.487 | 13 | 0.75±0.59 | 0.078 |
| • No | 40 | 0.35±0.25 | | 40 | 1.05±0.60 | |
| **Multiple species** | | | | | | |
| • Yes | 9 | 0.32±0.59 | 0.739 | 9 | 0.72±0.63 | 0.073 |
| • No | 44 | 0.35±0.26 | | 44 | 1.03±0.59 | |

Mann-Whitney U test

*Statistically significant difference (p<0.05)

*Candida* carriers (0.27±0.23 and 0.77±0.56 ml/min, respectively) were significantly lower than those of non-*Candida* carriers (0.40±0.27 and 1.16±0.59 ml/min, p = 0.042 and 0.007, respectively). In addition, mean SFR was lower in participants who used xerostomic drugs (0.89 ±0.61 ml/min) than those who did not (1.15±0.56 ml/min), but the difference was only statistically marginally significant (p = 0.053). There was no statistically significant difference with regards to other factors examined.

## Discussion

In this study, we examined the relationship of hyposalivation to oral health indices and oral *Candida* carriage in a population of dentate Thai elders with no or well-controlled systemic conditions. We found that oral *Candida* colonization was higher in participants with hyposalivation both in univariate analysis and after adjusted for age. Hyposalivation was also associated with higher gingival and tongue-coating indices, but not root caries index. These two indices and the quantity of oral *Candida* load were also negatively correlated with salivary flow rates. Our findings indicate that hyposalivation is a major risk factor for poorer oral health and *Candida* colonization in independent dentate elders. Previous studies suggested that hyposalivation was associated with edentulism and in elders with fewer teeth, possibly due to reduced occlusal forces [8]. However, present day elders retain more natural teeth and edentulism has

declined [4], thus this study focused on elders with at least 4 pairs of posterior occlusal contacts.

We observed significant associations between hyposalivation and higher gingival and tongue-coating indices (Table 4, *p = 0.003* and *0.015*, respectively). Moreover, we also found that salivary flow rates negatively correlated with gingival and tongue-coating indices (Fig 1). However, we did not detect significant relationship with the root caries index. The reduced salivary flow could lead to reduced clearance and decreased immune components against oral microorganisms, which result in oral microbial dysbiosis, increased plaque accumulation and *Candida* adherence to the oral mucosa [10, 11, 32]. Thus, hyposalivation could promote gingival inflammation and adversely affect oral and systemic health; these are particularly important for the elders [10]. Furthermore, oral microorganisms could be transferred to the gut, and this transition was found to be higher in the elders, suggesting that gut microbiota and systemic health could be affected by oral microbiota [33]. In addition, we found that hyposalivation was significantly associated with objective dry mouth signs (Table 2, *p = 0.003*), but not subjective dry mouth symptoms (*p = 0.448*). Therefore, oral examination for dry mouth signs is important for identifying patients who require interventions to reduce the risk of these adverse effects of hyposalivation.

We observed oral *Candida* colonization in 47.2% of this elderly population, 76% and 52% of whom had *C. albicans* and non-*albicans Candida* species, respectively. The overall prevalence was similar to previous reports of 25.7%–55% *Candida* colonization in healthy population of various age groups, but the prevalence of NACS (24.5% of the population, 52% of *Candida* carriers) was relatively high when compared to 0%–30% in other studies in Thailand [21, 34–36]. The most frequently isolated NACS in this study was *C. glabrata*, followed by *C. dubliniensis*, *C. parapsilosis*, *C. krusei*, and only 1 case of *C. tropicalis* (Table 3). This finding differs from previous reports that suggested distinct geographical distribution of *Candida* species, where *C. glabrata* and *C. parapsilosis* were commonly detected in North America, while *C. tropicalis* was more prevalent in Asia-Pacific [19]. However, it has been suggested that age-related compromising conditions favoured *C. glabrata* colonization in elders [37]. A previous study in japanese community dwelling elders also reported that *C. albicans*, *C. glabrata*, *and C. dubliniensis* dominated the oral mycobiome [38]. Colonization by distinct species of *Candida* may have different effects on oral health. Interestingly, multi-species colonization by *C. albicans*, *C. glabrata*, *C. tropicalis*, and *C. krusei* was associated with atrophic mucosa in patients with xerostomia [39]. Furthermore, several species of NACS are intrinsically more resistant, or could frequently develop resistance, to the commonly used antifungal drugs, and may cause refractory candidiasis [40]. Since *Candida* colonized in the oral cavity could serve as a reservoir for oral and systemic infections when host immunity becomes compromised, the prevalence of oral carriage of *Candida*, especially of NACS, in the elderly is of concern [22]. Therefore, identification of risk factors associated with oral *Candida* colonization, especially of NACS, is important.

We found significantly higher prevalence of *Candida* colonization in the hyposalivation group (*p = 0.010;* adjusted OR = 4.36) and *Candida* carriage was associated with lower salivary flow rates (Table 8, *p = 0.042* and *0.007* for USFR and SFR, respectively). We also observed significant negative correlations between salivary flow rates and the quantity of *Candida* in the oral cavity (Fig 1A and 1B). These are consistent with previous reports that decreased salivary flow rate is a risk factor for *Candida* colonization [17, 32, 39, 41, 42]. Of note, patients with higher *Candida* counts were shown to have higher risk for candidiasis [22, 43, 44]. In addition, an animal study showed that *Candida* could induce bacterial dysbiosis that facilitates mucosal invasion and infection [45]. At the same time, high *Candida* load was also associated with low microbiome diversity that dominated by saccharolytic and acidogenic bacterial species in the

saliva of elders [46]. This suggests that conditions that favor high level of *Candida* carriage also affect other microorganisms that influence other aspects of oral health.

Interestingly, we observed a significant association between denture use and NACS colonization (Table 5, *p = 0.017*) This is consistent with our previous study in xerostomic post-radiotherapy Head and Neck cancer patients [17]. The use of denture was associated with a higher *Candida* colonization rate in mexican elderly women, with many isolates showing resistance to fluconazole [47]. *Candida* has the ability to form biofilm on the rough and porous surface of acrylic denture base [48]. Interestingly, we found *C. glabrata* as the most common NACS in this study and most (4 in 5 cases, 80%) were isolated from denture wearers. It has been suggested, based on *in vitro* studies and in denture stomatitis patients, that *C.glabrata* can form biofilm on denture surfaces and may have synergistic interactions with *C. albicans* [18]. Our study also found *C. glabrata* co-colonized with *C. albicans* (2 in 5 cases) and *C. dubliniensis* (1 in 5 cases). Thus, the interactions between species may also contribute to colonization, especially in denture wearers. Poor denture hygiene allows microbial accumulation and may result in mucosal inflammation and infection in denture stomatitis [16, 48]. Thus, appropriate denture cleaning protocols should be recommended and the use of non-toxic agent with antifungal activity may provide additional benefit [49]. We did not find statistically significant difference in denture plaque index of *Candida* carriers and non-carriers (Table 6). However, the number of participants who use dentures in the current study (N = 11) was too small to analyze the effect of denture hygiene on *Candida* colonization.

In patients with denture stomatitis, alterations in salivary proteome that may influence *Candida* colonization was observed [12, 18]. Increased expression of salivary proteins, such as cystatins and statherin, may facilitate biofilm formation, while increased in immunoglobulins and complement proteins may reflect immune responses to increased microbial burden [12]. In addition, changes in salivary secretion, such as in Sjögren syndrome, also affect the salivary proteome, especially on proteins related to innate immunity [50]. Thus, it would be interesting to further examine the changes in salivary proteome and how they affect *Candida* species colonization in elders with hyposalivation.

In this study, we investigated the effects of hyposalivation on oral health status and oral *Candida* carriage by focusing on independent dentate elderly population with no or well-controlled systemic diseases. This helps to minimize the effects of other potential confounding factors. Moreover, we examined both xerostomia and hyposalivation, and measured both unstimulated and stimulated salivary flow rates, so we could analyze all of these variables in relations to oral health and *Candida* colonization. Furthermore, to reduce examiner's bias, only one trained dentist and one scientist who were blinded to the salivation status of the participants performed clinical examination and microbiological analysis, respectively. In addition, the detection of NACS and multi-species colonization in this study was facilitated by the use of both chromogenic *Candida* agar and PCR for species identification to ensure accuracy. Nevertheless, this study still carries certain limitations. This is a cross-sectional study in a small group of participants. Thus, we cannot establish the time sequence of events. It would be interesting to examine the relationships in larger longitudinal studies and to identify effective interventions to mitigate the adverse effects of hyposalivation.

Within the limitations of this study, we conclude that hyposalivation is a risk factor for poorer oral health, especially in tongue coating and gingival inflammation, and oral *Candida* colonization in independent dentate elders. These could adversely affect their oral and systemic health, thus we suggest that hyposalivation be carefully monitored in the elders by observing objective dry mouth signs.

## Supporting information

**S1 File.**
(XLSX)

## Acknowledgments

The authors sincerely thank all participants in this study, the director and dental department staff of Phaholphonpayuhasena hospital for assistance in patient recruitment, Ms. Sureeporn Muangsawat for technical assistance and Assoc. Prof. Dr. Waranuch Pitiphat of Khon Kaen University for kind assistance on statistical analysis.

## Author Contributions

**Conceptualization:** Orapin Komin, Oranart Matangkasombut.

**Data curation:** Nada Buranarom, Oranart Matangkasombut.

**Formal analysis:** Nada Buranarom, Oranart Matangkasombut.

**Funding acquisition:** Oranart Matangkasombut.

**Investigation:** Nada Buranarom.

**Methodology:** Nada Buranarom, Orapin Komin, Oranart Matangkasombut.

**Project administration:** Nada Buranarom, Oranart Matangkasombut.

**Resources:** Orapin Komin, Oranart Matangkasombut.

**Supervision:** Orapin Komin, Oranart Matangkasombut.

**Visualization:** Nada Buranarom, Oranart Matangkasombut.

**Writing – original draft:** Nada Buranarom, Oranart Matangkasombut.

**Writing – review & editing:** Nada Buranarom, Orapin Komin, Oranart Matangkasombut.

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
