## [Decision Letter · Decision Letter 0]

14 Oct 2020

PONE-D-20-29925

Hyposalivation, oral health, and Candida colonization in independent dentate elders

PLOS ONE

Dear Dr. Matangkasombut,

Thank you for submitting your manuscript to PLOS ONE. After careful consideration, we feel that it has merit but does not fully meet PLOS ONE’s publication criteria as it currently stands. Therefore, we invite you to submit a revised version of the manuscript that addresses the points raised during the review process.

The reviewers were pleased with the manuscript however there are missing information in the introduction, methods and discussion. Please see comments from Reviewers and Editor.

We look forward to receiving your revised manuscript.

Kind regards,

Sompop Bencharit, DDS, MS, PhD, FACP

Academic Editor

PLOS ONE

Journal Requirements:

Additional Editor Comments:

Thank you for submitting the work to PLoS ONE. While the reviewers and editor see great potential of the study, there are several issues needed to be addressed:

1. In the introduction: The authors mentioned the changes of salivary proteomes based on changes in saliva flow. There is not sufficient discussion of which proteins that may be insufficient or more prevalent that would alter oral candidiasis. There are also a role of denture wearing and candidal species needed to be discussed. The following studies may need to be reviewed and added to the introduction. There was also an interaction between different candidal species associated with denture stomatitis and denture wearing. For example the C. grabata species seems to associate with C. albicans based on proteomic study and there is a reduction in protective proteins in saliva. Please add more of your thought and hypothesis in the introduction based on current literature (see below).

Bencharit S, Altarawneh SK, Baxter SS, Carlson J, Ross GF, Border MB, Mack CR, Byrd WC, Dibble CF, Barros S, Loewy Z, Offenbacher S. Elucidating role of salivary proteins in denture stomatitis using a proteomic approach. Mol Biosyst. 2012 Oct 30;8(12):3216-23. doi: 10.1039/c2mb25283j. PMID: 23041753; PMCID: PMC3519238.

Xie H, Onsongo G, Popko J, de Jong EP, Cao J, Carlis JV, Griffin RJ, Rhodus NL, Griffin TJ. Proteomics analysis of cells in whole saliva from oral cancer patients via value-added three-dimensional peptide fractionation and tandem mass spectrometry. Mol Cell Proteomics. 2008 Mar;7(3):486-98. doi: 10.1074/mcp.M700146-MCP200. Epub 2007 Nov 28. PMID: 18045803.

Aqrawi LA, Galtung HK, Vestad B, Øvstebø R, Thiede B, Rusthen S, Young A, Guerreiro EM, Utheim TP, Chen X, Utheim ØA, Palm Ø, Jensen JL. Identification of potential saliva and tear biomarkers in primary Sjögren's syndrome, utilising the extraction of extracellular vesicles and proteomics analysis. Arthritis Res Ther. 2017 Jan 25;19(1):14. doi: 10.1186/s13075-017-1228-x. PMID: 28122643; PMCID: PMC5264463.

Javed F, Al-Kheraif AA, Kellesarian SV, Vohra F, Romanos GE. Oral Candida carriage and species prevalence in denture stomatitis patients with and without diabetes. J Biol Regul Homeost Agents. 2017 Apr-Jun;31(2):343-346. PMID: 28685534.

Yarborough A, Cooper L, Duqum I, Mendonça G, McGraw K, Stoner L. Evidence Regarding the Treatment of Denture Stomatitis. J Prosthodont. 2016 Jun;25(4):288-301. doi: 10.1111/jopr.12454. Epub 2016 Apr 6. PMID: 27062660.

Byrd WC, Schwartz-Baxter S, Carlson J, Barros S, Offenbacher S, Bencharit S. Role of salivary and candidal proteins in denture stomatitis: an exploratory proteomic analysis. Mol Biosyst. 2014 Jul 29;10(9):2299-304. doi: 10.1039/c4mb00185k. PMID: 24947908; PMCID: PMC4119604.

2. In the Methods, please add a table for detailed inclusion and exclusion criteria. The presence of denture should be addressed with denture hygiene and habit, e.g. night denture wearing as suggested by Reviewer #2. Have you documented the denture wearing and hygiene habit? Do you have data on edentulous patients or how many teeth remaining? Please add these data.

3. In the Discussion, please discuss the interaction between the most common non albican species, C. glabata with C. albicans and see if this ties back into your introduction. Please also discuss the denture hygiene with oral candidiasis.

Reviewers' comments:

Reviewer's Responses to Questions

**Comments to the Author**

1. Is the manuscript technically sound, and do the data support the conclusions?

Reviewer #1: Yes

Reviewer #2: Partly

2. Has the statistical analysis been performed appropriately and rigorously? 

Reviewer #1: I Don't Know

Reviewer #2: Yes

3. Have the authors made all data underlying the findings in their manuscript fully available?

Reviewer #1: Yes

Reviewer #2: Yes

4. Is the manuscript presented in an intelligible fashion and written in standard English?

Reviewer #1: Yes

Reviewer #2: Yes

5. Review Comments to the Author

Reviewer #1: I found it interesting that you chose to exclude subjects with oral candidaisis, as this is an important potential outcome of oral Candida species colonization. Overall, a well set-out cross sectional study. The data appears to support the conclusions.

Reviewer #2: Methods should include a description of (1) how denture wearers care for their dentures and (2) any treatment by the study participants for hyposalivation (Biotene). A major gap in the study is the lack of information regarding oral care administered by the study participants.

6. PLOS authors have the option to publish the peer review history of their article (what does this mean?). If published, this will include your full peer review and any attached files.

Reviewer #1: No

Reviewer #2: No

---

## [Author Response · Author response to Decision Letter 0]

26 Oct 2020

Dear editor, 

We sincerely thank the reviewers and editor for useful comments and suggestions that helped to improve our manuscript. The revised texts are highlighted in yellow in the manuscript and our responses to the comments are listed in blue italic fonts below in a point-by-point fashion. We hope that the revision satisfies the requirements of the journal and the editor and that the manuscript is now in a suitable form for publication in PLOS One. 

Point-by-point responses to reviewers and editor’s comments:

Journal Requirements:

Response:- We have prepared the revised manuscript according to the instructions provided. 

Response:- We have added the data into the manuscript (Table 6) and deleted the phrase “data not shown”. All data are provided in the supplementary data file.

Additional Editor Comments:

Thank you for submitting the work to PLoS ONE. While the reviewers and editor see great potential of the study, there are several issues needed to be addressed:

1. In the introduction: The authors mentioned the changes of salivary proteomes based on changes in saliva flow. There is not sufficient discussion of which proteins that may be insufficient or more prevalent that would alter oral candidiasis. There are also a role of denture wearing and candidal species needed to be discussed. The following studies may need to be reviewed and added to the introduction. There was also an interaction between different candidal species associated with denture stomatitis and denture wearing. For example, the C. glabrata species seems to associate with C. albicans based on proteomic study and there is a reduction in protective proteins in saliva. Please add more of your thought and hypothesis in the introduction based on current literature (see below).

Response:- We thank the editor for the useful suggestions. We have included additional information together with necessary references on the changes in salivary proteome, Candida species interaction, and denture wearing in the introduction and/or discussion as suggested. (Page 4, line 74-77; Page 5, line 93-94; Page 28, line 432-439; Page 29, line 446-453)

2. In the Methods, please add a table for detailed inclusion and exclusion criteria. The presence of denture should be addressed with denture hygiene and habit, e.g. night denture wearing as suggested by Reviewer #2. Have you documented the denture wearing and hygiene habit? Do you have data on edentulous patients or how many teeth remaining? Please add these data.

Response:- We have now added a table describing the inclusion and exclusion criteria (Table 1 page 6), and information on denture plaque index (page 9 line 169-177, and Table 6 Page 21) and denture hygiene practice (page 20, line 314-321). Because we aimed to focus on only dentate subjects in this study, there was no edentulous patients. We have also added the information on number of remaining teeth in Table 2 (page 13). 

3. In the Discussion, please discuss the interaction between the most common non albican species, C. glabrata with C. albicans and see if this ties back into your introduction. Please also discuss the denture hygiene with oral candidiasis.

Response:- We have added further discussion on the interaction between C.glabrata and C.albicans and denture hygiene as suggested. (Page 28, line 432-439; Page 29, line 446-453; and Page 20, line 314-321)

Reviewers' comments:

Reviewer's Responses to Questions

Comments to the Author

5. Review Comments to the Author

Reviewer #1: I found it interesting that you chose to exclude subjects with oral candidiasis, as this is an important potential outcome of oral Candida species colonization. Overall, a well set-out cross sectional study. The data appears to support the conclusions.

Response:- We thank the reviewer for the positive comments. The reason that we exclude patients with oral candidiasis was because, in this study, we focused on the relationship between hyposalivation, oral health status, and Candida colonization in relatively healthy independent elders. In addition, we aimed to examine Candida carriage in the absence of oral infection. Thus, we did not include patients with oral candidiasis also because it may have resulted from systemic immunocompromising conditions. 

Reviewer #2: Methods should include a description of (1) how denture wearers care for their dentures and (2) any treatment by the study participants for hyposalivation (Biotene). A major gap in the study is the lack of information regarding oral care administered by the study participants.

Response:- We thank the reviewer for the useful suggestions. We have now added the information on denture hygiene practice of denture wearers (page 20, line 314-321) and denture plaque index (Table 6). To ease their dry mouth symptoms, most of the subjects with xerostomia in this study take frequent sips of water (page 12, line 232-234). Furthermore, Biotene is not available in Thailand. 

Should you have any further questions or suggestions, please contact me at Oranart.m@chula.ac.th. 

Best regards,

Oranart Matangkasombut

Orapin Komin

---

## [Editor Report · Decision Letter 1]

10 Nov 2020

Hyposalivation, oral health, and Candida colonization in independent dentate elders

PONE-D-20-29925R1

Dear Dr. Matangkasombut,

We’re pleased to inform you that your manuscript has been judged scientifically suitable for publication and will be formally accepted for publication once it meets all outstanding technical requirements.

Kind regards,

Sompop Bencharit, DDS, MS, PhD, FACP

Academic Editor

PLOS ONE

Additional Editor Comments (optional):

Thank you for your revision. The authors have sufficiently addressed all comments and revised the manuscript according to the comments from the reviewers and editor.

---

## [Editor Report · Acceptance letter]

14 Nov 2020

PONE-D-20-29925R1 

Hyposalivation, oral health, and *Candida* colonization in independent dentate elders 

Dear Dr. Matangkasombut:

I'm pleased to inform you that your manuscript has been deemed suitable for publication in PLOS ONE. Congratulations! Your manuscript is now with our production department. 

Kind regards, 

on behalf of

Dr. Sompop Bencharit 

Academic Editor

PLOS ONE